# International Perspectives on the Implications of Cannabis Legalization: A Systematic Review & Thematic Analysis

**DOI:** 10.3390/ijerph16173095

**Published:** 2019-08-26

**Authors:** Anees Bahji, Callum Stephenson

**Affiliations:** 1Department of Public Health Sciences, Queen’s University, Kingston, ON K7L 3N6, Canada; 2Department of Psychiatry, Queen’s University, Kingston, ON K7L 3N6, Canada; 3School of Kinesiology and Health Studies, Queen’s University, Kingston, ON K7L 3N6, Canada

**Keywords:** cannabis, legalization, marijuana

## Abstract

The legality, recreational and medical use of cannabis varies widely by country and region but remains largely prohibited internationally. In October 2018, Canada legalized the recreational use of cannabis—a move many viewed as controversial. Proponents of legalization have emphasized the potential to eradicate the marijuana black market, improve quality and safety control, increase tax revenues, improve the availability of medical cannabis, and lower gang-related drug violence. Conversely, opponents of legalization have stressed concerns about cannabis’ addictive potential, second-hand cannabis exposure, potential exacerbation of underlying and established mental illnesses, as well as alterations in perception that affect safety, particularly driving. This systematic review synthesizes recent international literature on the clinical and public health implications of cannabis legalization.

## 1. Introduction

Globally, cannabis is the most widely used illicit drug [1], and cannabis use and dependence are estimated to have increased over the past two decades [2]. In 2016, the United Nations Office on Drugs and Crime estimated 192 million people aged 15–64 years used cannabis [3]. The Global Burden of Disease study estimated the age-standardized rate of cannabis use disorder in 2017 was 289.7 per 100,000 population (95% Uncertainty Interval (UI) 248.9–339.1), affecting 22.1 million people (95% UI 19.0–25.9 million) [2]. The United States and Canada are estimated to have among the highest age-standardized rates of cannabis use disorders (CUDs) in the world [2]. In Canada, the prevalence of cannabis use and CUDs have increased over the past decade, paralleling changes in the legal and political climate favoring legalization [4,5]. In 2017, 4.5 million Canadians aged 15 years or older reported past-year cannabis use, 1.5 million reported daily or near-daily use, and 6.8% met criteria for CUD [6,7].

People who use cannabinoids may seek varied effects that include relaxation, euphoria, relief from stress, increased appetite, improved sleep, and self-confidence [8]. However, a range of adverse physical and psychological consequences may also be experienced. The effects of short-term use include impaired short-term memory, impaired motor coordination, altered judgment, and—at higher doses—paranoia and psychoses [9,10]. The effects of long-term or heavy cannabinoid use include addiction, altered brain development, poor educational outcome, cognitive impairment, diminished life satisfaction and achievement, symptoms of chronic bronchitis, and increased risk of chronic psychotic disorders [9]. There are also potentially fatal harms, particularly among people who use cannabinoids regularly or who are dependent. These include risks of injuries (both unintentional and intentional, including exposure to violence), motor vehicle collisions, and suicide [9,11].

In recent years, recreational cannabis use has become increasingly decriminalized and legalized in many jurisdictions, including Canada in October 2018 [12]. Cannabis has also been legalized for recreational (and often medicinal use) in many U.S. states—including Colorado and California—while remaining a Schedule-I drug federally [13]. Despite this, illicit sales and use of cannabis continue to be prevalent, particularly among marginalized people who use illicit drugs [14].

Proponents of legalization have emphasized the potential to eradicate the marijuana black market [15], improve quality and safety control [15], increase tax revenues [16], improve the availability of medical cannabis [17] and lower gang-related drug violence [18]. As legalization regimes are established in multiple countries, public health professionals are increasingly synthesizing decades of knowledge from other policy areas to inform effective cannabis policy [13]. Conversely, opponents of legalization have stressed concerns about cannabis’ addictive potential [1], second-hand cannabis exposure [19], potential exacerbation of underlying and established mental illnesses [20], as well as alterations in perception and attitudes towards cannabis, particularly those that affect safety and driving.

Given these seemingly diametrically opposed views, the impact of legalization remains unclear. This is compounded by the fact that cannabis use is only legal in a handful of jurisdictions—including the Netherlands, Uruguay, Canada, and specific U.S. states. However, recent research involving cannabis has been increasing, and, studies measuring changes in cannabis-related outcomes before and after legalization are also on the rise. To date, there have been no reviews exploring the implications of cannabis legalization in jurisdictions where medical or recreational marijuana use is permitted.

The purpose of this review was to synthesize recent literature on this theme.

## 2. Methods

### 2.1. Reporting

We conducted a systematic review in accordance with the PRISMA guidelines [21].

### 2.2. Search Strategy

We searched three online databases (MEDLINE, EMBASE, and PsycINFO) for recent articles (published since 2018) exploring the clinical and public health implications of cannabis legalization.

### 2.3. Definition of Legalization

For the purposes of this review, a liberal definition of cannabis legalization as adopted, which we defined as legislation that permits either the medical or recreational use of cannabis in a defined geographic region [22]. A full-search strategy is provided in Appendix A.1. 

### 2.4. Eligibility Criteria

Articles exploring one or more implications of cannabis legalization were considered eligible, such as articles measuring the epidemiology of mental health conditions, health service utilization, rates of cannabis consumption. Articles were not excluded on the basis of study design, population considered, or geographic region, however, the review was restricted to human studies and to English language articles.

### 2.5. Study Selection

Citations were imported into Covidence—an online systematic review software—which facilitated the removal of duplicate citations [23]. Studies were screened by title and abstract by one author (Anees Bahji) and reviewed for consistency by a second (Callum Stephenson). Full-text articles were then reviewed by both authors for inclusion.

### 2.6. Data Extraction and Synthesis

Data were pooled by way of a qualitative narrative synthesis. Findings from individual studies were then grouped using thematic analysis into broad categories to facilitate meaningful discussion points. The capacity for a quantitative meta-analysis was limited due to the diverse nature of studies considered and outcomes reported.

## 3. Results

### 3.1. Systematic Review

A total of 36 studies met inclusion criteria for this systematic review (Appendix A.2), covering a variety of themes: prevalence and trends in cannabis consumption (n = 5); health implications (n = 9); healthcare utilization (n = 7); criminality (n = 4); cannabis black-market implications (n = 9); and health policy (n = 2).

### 3.2. Cannabis Consumption Implications

Although only a few studies measured the prevalence of cannabis use following legalization, most found that the prevalence increased. For example, overall marijuana consumption increased in Washington [24], while rates of marijuana use by undergraduate students increased substantially following legalization of recreational cannabis in Colorado [25,26,27]. Conversely, marijuana legalization was found to increase the prevalence of cardiovascular complications and cardiac-related deaths [28].

### 3.3. Health Implications

Following medical marijuana legalization in several U.S. states, there were significant increases cardiac mortality rates [29], but there were concurrent reductions in the rates of opioid prescribing, particularly in areas where cannabis dispensaries were legal. However, in these states, there was a concurrent increase in tobacco sales [30]. Compliance rates among chronic pain patients who were treated with opioids did not change following legalization in select U.S. states [31]. In areas where medical marijuana was legalized, the prevalence of serious mental illnesses—like schizophrenia and bipolar disorder—were significantly higher following legalization compared to the period before legalization [32]; however, these studies were not able to distinguish between true epidemiological rises in prevalence from increasing rates of diagnosis. While frequent cannabis use was associated with impulsivity at an individual level, there was no relationship observed between population-based cannabis use and rates of impulsive behaviour [33].

### 3.4. Public Opinions Towards Cannabis Legalization

In Uruguay, public views towards cannabis liberalisation were intertwined with concerns about public security and apprehension that it will open the gate to heavier drugs (like amphetamines or opioids) rather than with concerns about individual health and demographic factors [34].

### 3.5. Maternal/Infant/Child Health Implications

Available data suggests that cannabis use during pregnancy is relatively common and persistent, despite knowledge of the potential risks of harm [35]. However, views and perspectives toward legalization vary among pregnant women and may impact cannabis use during pregnancy [36]. Similarly to the general population, rates of marijuana use have increased after marijuana legalization among pregnant and parenting women in Washington [37]. However, there were no changes in the prevalence of low birth weight or small for gestational age births during this same interval [38]. Despite these findings, the risks of prenatal cannabis exposure should be neither overstated nor minimized, and that the legal status of a substance should not be equated with safety [39]. Scientifically accurate dissemination of cannabis outcomes data is necessary [40], and clinicians must recognize that even in environments where cannabis is legal, pregnant women may end up involved with children’s protective services [41].

### 3.6. Healthcare Utilization Implications

Retrospective studies have observed notable increases in the number and rates of emergency department (ED) visits for cannabis-related presentations after recreational marijuana legalization in the United States [42]. Likewise, the prevalence of psychiatric comorbidity is substantially higher in adults presenting to the ED for cannabis-associated diagnostic codes than for other visits [43]. Rates of hospitalization (32.9% vs. 18.9%) and the ED length stay (3 vs. 2 hours) tend to be higher for visits involving inhaled cannabis than edible cannabis [44]. These differences might indicate greater severity of adverse events with inhalable cannabis, but they also may also reflect differences in the underlying clinical presentation [44].

### 3.7. Criminal Implications

In a retrospective analysis comparing crime data in neighbouring American states, Washington and Oregon, revealed that reported rape, property crimes, and thefts all decreased by 15–30%, 10–20%, and 13–22% respectively after legalization in Washington [24]. Moreover, the prevalence of these crimes occurring across the border where recreational cannabis use is illegal, in Oregon, remained the same. Furthermore, in 2018, all divisions of police clearance rates either remained stagnant or improved in Colorado and Washington following recreational marijuana legalization [45]. Marijuana-related arrests in Washington decreased from 5,531 in 2012 to 120 in 2013, allowing for more police resource allocation to other divisions [45]. However, there was no alteration in the racial disparity of marijuana related arrests after legalization, with African Americans still accounting for 2.7 times more arrests than Caucasians [46].

### 3.8. Cannabis Black Market Implications

Although legalization was intended to reduce illicit cannabis sales, the black market for cannabis in Canada has actually increased with legalization—not decreased [47]. This has occurred largely because more marijuana is available from legal sources to sell illegally [48]. According to preliminary reports from Statistics Canada, 79% of cannabis was bought illegally in the fourth quarter of 2018, down from 90% in the third quarter [49]. Recently released reports indicate that Canadians buying cannabis from legal sources pay about $10 per gram, while those who have stuck with the grey market pay $6.37 per gram [49].

U.S. states without legalized marijuana bordering states with legalized marijuana are thought to have increased availability of marijuana for the black market [50,51]. For example, despite legalizing cannabis in 2016, California still has a thriving black market, with as much as 80% of all cannabis sales being linked to illegal sources [52,53]. Recent economic estimates suggest that California’s illicit cannabis market is worth approximately $3.7 billion—more than four times the size of the legal market in the state [54,55].

### 3.9. Policy Implications

One study reported a risk–benefit framework on the impact of legalization on mental disorders—drawing on the impact of cannabis use on incidence, prevalence and severity of mental disorders [56]. Similarly, the Canadian Psychiatric Association released a position statement outlining potential concerns on the implications of legalization on mental health [40], highlighting concerns about early exposure and abnormal brain development.

## 4. Discussion

### 4.1. Summary of Findings

This review identified 36 studies exploring diverse aspects of cannabis legalization—health, epidemiology, health service utilization, public policy, crime, and economic implications. Although only a few studies measured the prevalence of cannabis use following legalization, most found that the prevalence of cannabis use increased over time. Available data also suggests that cannabis use during pregnancy is relatively common and persistent, despite knowledge of the potential risks of harm—however, the long-term effects of cannabis exposure in utero remains unclear. Legalization of marijuana in the regions of the United States was followed by increases in the number and rates of emergency department visits for cannabis-related presentations (such as cannabis intoxication and cannabis-related hyperemesis). From an economic perspective, early studies indicate that legalization—which was intended to reduce illicit cannabis sales—may have created a nexus for a stronger illicit black market for cannabis sales in Canada, which have actually increased post-legalization. Still, cannabis-related criminal activity has reduced in certain regions where cannabis has been legalized.

### 4.2. Significance of Findings

Collectively, the effects of cannabis legalization are incredibly heterogeneous, however, research is beginning to demystify certain beliefs about cannabis in the era of legalization. As such, research involving the implications of cannabis legalization is also increasing at a rapid rate. However, there is still significant controversy regarding the overall impact of legalization—particularly on mental health and public policy. As such, cannabis policy is rapidly evolving in Canada, the United States, and the rest of the world as more jurisdictions legalize medical and recreational marijuana use.

Overall, public opinion has shifted dramatically in favor of marijuana legalization, particularly in the United States. Cannabis use is also on the rise even among older adults, who have historically been left out of discussions pertaining the cannabis and drug policy in general. Strikingly, the increasing prevalence of cannabis use occurs in the background of social perceptions that cannabis is associated with a relatively low associated risk, which diametrically opposes current knowledge about the biological and clinical effects of cannabis use in both the short and long term [57].

### 4.3. Limitations

Although this review has notable strengths, the findings presented here should be interpreted in light of some significant limitations. First, as a systematic review, the quality and availability of the evidence presented is limited by the extent of published literature—this is particularly relevant given that extent of cannabis legalization worldwide is fairly limited, which precludes extensive research on this topic. Second, as this review focused primarily on recent literature (published since 2017), several key articles that were published prior to the search date were excluded from the discussion. As such, the contributions of research articles not discussed in this review should not be discounted because this review did not mention them. For example, there is a much larger well of knowledge on cannabis legalization and criminal activity than the selection of articles covered by this review. Third, given the scope of this topic, a formal quantitative meta-analysis was not conducted—this may be undertaken in future studies focusing on specific aspects of cannabis legalization (such as the prevalence of specific emergency room visits). Fourth, our review excluded non-English language studies, which may have precluded the inclusion of key articles from non-English speaking jurisdictions, such as Uruguay.

## 5. Conclusions

Despite changing legal climates, there is an increasing demand for clear and consistent messaging on the effects of cannabis use. Currently, there is a paucity of literature on a variety of implications related to cannabis legalization—and the available studies are fairly heterogeneous in their findings. As such, clear conclusions are difficult to draw at this point in time. However, with legalization and liberalization of cannabis underway in many parts of the world, it is likely that the answers to these questions will become available in the near future. Thus, the ongoing accumulation of empirical data will be helpful to inform ongoing debates about the role of public policy on cannabis legislation.

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
