# Peer review of "International Perspectives on the Implications of Cannabis Legalization: A Systematic Review & Thematic Analysis"

_ijerph, 2019, doi:10.3390/ijerph16173095_

Round 1

Reviewer 1 Report

The authors revisions have addressed many of the reviewers' comments. A few additional points relevant to this revision

It isn't clear why the authors have restricted their search to studies after 2017. Some of the larger literature on cannabis and traffic safety and on crime is lost.  The prominent NASEM report on cannabis (admittedly health effects, not effects of legalization) is worth citing in the introduction for the overview it gives of the science.. Analogous Canadian publicatiosn would also be useful to know about. Restriction to English language articles - noted in 2.4/ line 65: authors might wish to remind readers of this in the abstract and/or limitations, especially since many jurisdictions thinking of legalizing cannabis are outside the English-speaking world (e.g. Mexico/ future possible legalization) Section 3.2/ lines 86-87: was the Abouk study referenced (25) about workplace safety or about cardiac deaths in older individuals? Please check reference and the conclusion which seems to suggest that marijuana use was protective - might wish to elaborate if that was indeed a correct characterizatio of the finding. Section 3.3/ line 94-95: what does it mean that prevalence of mental illnesses was higher in areas with legalization - is this before legalization? After legalization? A function of diagnosis ("diagnosed more frequently" may be different from higher prevalence)? Section 3.3/ line 97-100: Uruguay study seems to be about opinions rather than health implications. It is not clear why the authors are mixing public opinion and actual health implications - this study likely should be placed elsewhere (eg. under a category of public opinion on legalization) or dropped. There is a much larger literature on crime and marijuana than the studies in 3.6. Restricting to studies published after 2018 seems to 

Author Response

Thanks for this feedback. We’ve created a limitations sections and have included this as a key limitation. Section 3.2/ lines 86-87: was the Abouk study referenced (25) about workplace safety or about cardiac deaths in older individuals? Please check reference and the conclusion which seems to suggest that marijuana use was protective - might wish to elaborate if that was indeed a correct characterization of the finding. Thanks for this feedback. You are correct—the Abouk study was about cardiac deaths following legalization. We have modified the sentence to correct the conclusion that we had originally reported. Section 3.3/ line 94-95: what does it mean that prevalence of mental illnesses was higher in areas with legalization - is this before legalization? After legalization? A function of diagnosis ("diagnosed more frequently" may be different from higher prevalence)? Thanks for this feedback. This is a major limitation of these studies. We have revised the sentence to read as follows. “In areas where medical marijuana was legalized, the prevalence of serious mental illnesses—like schizophrenia and bipolar disorder—were significantly higher following legalization compared to the period before legalization [32]; however, these studies were not able to distinguish between true epidemiological rises in prevalence from increasing rates of diagnosis.” Section 3.3/ line 97-100: Uruguay study seems to be about opinions rather than health implications. It is not clear why the authors are mixing public opinion and actual health implications - this study likely should be placed elsewhere (eg. under a category of public opinion on legalization) or dropped. Thanks for this feedback—and for catching this oversight. We have moved this portion to create a new section called “Public Opinion on Legalization”. There is a much larger literature on crime and marijuana than the studies in 3.6. Thanks for this feedback. You are correct—however, our study’s limitation of studies published since 2017 limited the array of articles that our study discussed.

Reviewer 2 Report

The manuscript looks good to go as a brief report, with the possible addition of a stronger paragraph in the Discussion, pointing to primary observations in the review. The Discussion appears quite brief. What, overall, do we take away from this analysis?

Author Response

Reviewer 2

The manuscript looks good to go as a brief report, with the possible addition of a stronger paragraph in the Discussion, pointing to primary observations in the review. The Discussion appears quite brief. What, overall, do we take away from this analysis? Thanks for this feedback. We have updated the discussion and divided it into a few sections. First, we presented a summary of the findings. Second, we discussed the significance of the findings from the perspective of the authors. Third, we added a section discussing the limitations of the review process. Fourth, we added a conclusion as a final take-away section. We hope this revised discussion enhances the impact of the manuscript as a whole.

Round 2

Reviewer 1 Report

Reviewer comments were incorporated in this revision.